

# The relationship between anti-Müllerian hormone (AMH) levels and pregnancy outcomes in patients undergoing assisted reproductive techniques (ART)

Shalini Umarsingh[*], Jamila Khatoon Adam and Suresh Babu Naidu Krishna[*]

Biomedical and Clinical Technology, Faculty of Health Sciences, Durban University of Technology, Durban, South Africa

[*] These authors contributed equally to this work.

## ABSTRACT

A variety of predictors are available for ovarian stimulation cycles in assisted reproductive technology (ART) forecasting ovarian response and reproductive outcome in women including biomarkers such as anti- Müllerian hormone (AMH). The aim of our present study was to compare the relationship between AMH levels and pregnancy outcomes in patients undergoing intra-cytoplasmic sperm injection (ICSI). Overall, fifty patients ($n = 50$), aged 20–45 years were recruited for the present prospective study. Three AMH levels were presented with high often poly cystic ovarian syndrome (PCOS) amongst 52.4% patients, 40.5% in normal and 7.1% in low to normal, correspondingly. There was statistically significant relationship between AMH and day of embryo transfer ($p < 0.05$). The Pearson analysis between AMH, age, E2 and FSH displayed no statistically significant relationship between E2 and AMH ($p < 0.05$) and negative correlation between FSH and age ($p > 0.05$). The area under the receiver operating characteristic curve for $E_2$ was 0.725 and for AMH levels as predictors of CPR was 0.497 indicating $E_2$ as better predictor than AMH. The number of oocytes, mature oocytes and fertilized oocytes all presented a weak positive relationship to AMH. Our results confirm the clinical significance of AMH to accurately predict ovarian reserve as a marker and its limitations to use as predictor for a positive pregnancy outcome. Additional prospective studies should be conducted to validate the predictive capability of AMH levels for the outcome of clinical pregnancy.

## INTRODUCTION

Couples in modern societies postpone childbearing amidst busy schedules and career advancements; trying to conceive at a more advanced age contributing to a rise in the occurrence of infertility (*Caroppo et al., 2006*). Most women are unaware that fertility starts to decline after the early thirties in some individuals. With increasing female age, fecundity in natural and stimulated ovarian cycles declines, as observed in population-based studies (*Grynnerup, Lindhard & Sorensen, 2012*) as well as in IVF studies (*Scheffer et al.,*

Corresponding author
Suresh Babu Naidu Krishna,
Sureshk@dut.ac.za

*2018*). For this reason, there is a growing number of women of advanced age seeking treatment for infertility (*Oskayli et al., 2019*).

The hormonal control of ovarian function is influenced by administering exogenous follicle stimulating hormone (FSH) (*Richards, 2018*). Prediction of ovarian responses prior to stimulation is not only useful for patient counselling, but also important in tailoring the optimal dosage of gonadotrophin for individual patients. The recruitment and development of multiple follicles in response to gonadotrophin stimulation are essential for the successful treatment of infertility by assisted reproductive techniques (ART) (*Dewailly et al., 2014*; *Yang, Wu & Zhang, 2020*). Besides, poor ovarian response has been suggested to be associated with high cycle cancellation rates (*Saldeen, Källen & Sundström, 2007*). *Chang et al. (1998)* found that patients with antral follicle number $\leq 3$ had a significantly higher rate of cycle cancellation and higher human menopausal gonadotropin (HMG) dosage as compared with those patients with antral follicle number 4–10 or $\geq 10$. Nevertheless, the AFC is presently believed to be the finest specific predictor of ovarian response to stimulation in ART, and it can be used in clinical practice for pretreatment counselling targets.

In assisted reproduction, serum levels for several hormones are used to assess the ovarian reserve and to monitor the development of the follicles that have been stimulated by gonadotrophins (*Alson et al., 2018*). Traditional techniques used to predict ovarian stimulation have incorporated serum levels of hormones such as FSH, LH and estrogen ($E_2$) along with ultrasonographic guides such as ovarian volume and number of early antral follicles as a reliable predictor of the outcome of in vitro fertilization (IVF) (*Kunt et al., 2011*). Over the last few years, the anti-Müllerian hormone (AMH) has been projected as a novel marker for predicting ovarian response to gonadotrophin stimulation (*Alson et al., 2018*; *Zargar, Najafian & Zamanpour, 2018*). AMH is a dimeric glycoprotein strongly produced by the granulosa cells of the pre-antral (primary and secondary) and small antral follicles (AF's) in the ovary and shown to be age dependent (*Sahmay et al., 2014*). Measurement of anti-Müllerian hormone in serum is much more precise measure of the ovarian reserve than the other hormones that have previously been available to us (*Anderson, Nelson & Wallace, 2012*).

The aim of this prospective study was to investigate the relationship between Anti-Müllerian hormone levels and pregnancy outcomes in patients undergoing *in-vitro* fertilization or intra-cytoplasmic sperm injection (ICSI).

## MATERIALS AND METHODS

### Patients
Fifty women ($n = 50$), aged 20–45 years were recruited from Centre of Assisted Reproduction and Endocrinology (C.A.R.E) Clinic in Westville, Durban, South Africa who were undergoing IVF treatment. This study was approved by Ethical Committee of the Durban University of Technology (Project reference 128/16) and Research Committee, C.A.R.E. Clinic, Durban, South Africa. After approving the study by the research ethics committees, written informed consents were obtained from all the patients.

## GnRH antagonist protocol

A gonadotrophin-releasing hormone (GnRH) antagonist protocol with recombinant FSH (GONAL-f, Merck Serono, Darmstadt, Germany) was used as downregulatory (*Park et al., 2015*). The second approach was followed by administering 0.25 mg/day Cetrotide (Merck Serono). When at least 3 or more follicles reach a diameter equal or above 17–18 mm, the endometrial thickness reached at least seven mm by ultrasound and E2 levels were about 1,500–1,800 pmol/L then Human chorionic gonadotropin (hCG) was administered. All patients received 5,000–10,000 IU hCG (Ovitrelle®, Merck Serono). Oocyte retrieval was performed 36 h after the administration of the hCG. Conventional ICSI was performed according to previously described protocols.

## Sample collection

Blood samples were collected every 3–4 days on commencement of the treatment. The blood samples were centrifuged at 3,000 rpm for 10 min using a Biofuge centrifuge (Biofuge Primo–Heraeus) to obtain the blood serum. AMH and FSH levels were recorded, upon the first visit. Estrogen($E_2$) and LH levels were monitored throughout the program until a peak $E_2$ and LH level were reached.

## Hormone assays

Gen II ELISA (Beckman Coulter Inc., USA, catalog number A79765/A79766, unmodified version). (Beckman Coulter, USA) kit was used to estimate hormone levels (FSH-Cat. No. 33520 Access hFSH reagent, 100 determinations, $2 \times 50$ tests); E2 (Cat. No. B84493 Access Sensitive Estradiol Reagent, 100 determinations) and AMH (Cat. No. B13127 Access AMH Reagent, 100 determinations, $2 \times 50$ tests) from the blood serum according to manufacturer's instructions. Insemination and intra cytoplasmic sperm injection (ICSI), oocyte retrieval, culture, fertilization, embryo culture, and transfer were carried out as previously described by *Gardner et al. (2001)*.

## Inclusion criteria

The population of the study included female patients ranging between the ages of 20–45.

## Exclusion criteria

Patients undergoing cancer therapy and patients on immune suppressant drugs were excluded from study.

## Statistical analysis

The data were analysed using IBM SPSS software (Chicago, IL, USA). Pearson's correlation was used to determine the direction, strength, and significance of the correlation between X and Y variables between the different semen parameters. A parametric multiple linear regression analysis was used to evaluate the relationship between AMH and other available endocrine markers. ROC curves were used to assess predictive value for $E_2$ and AMH and evaluating cut off values to optimise sensitivity and specificity. A *p* value of $< 0.05$ was considered statistically significant.

## Institutional Review Board approval

This study was approved by Ethical Committee of the Durban University of Technology (Project reference 128/16) and Research Committee, C.A.R.E. Clinic, Durban, South Africa and was performed in accordance with the Helsinki Declaration of 1975 (as revised in 1983).

## RESULTS

The prospective study included fifty patients who met the inclusion criteria. From the initial sample size of fifty, forty-two presented with data that could be analysed whilst 8 patients had oocytes that where abnormal and did not result in transfer. The data from these 8 patients were not included in the study due to poor embryo development (Table 1).

Amongst the 42 patients analysed, 4.76% were between 20–24 years, 9.52% were between 25–29 years, 40.47% were between 30–34 years, 35.7% were between 35–39 years and 9.52% were between 40–44 years, respectively. As demonstrated by this study the clinical pregnancy rate for patients 20–24 years was 100%, 25–29 years was 50%, 30–34 years was 17.6%, 35–39 years was 26.6% and 40–44 years was 25% (Fig. 1).

Table 2 shows number of oocytes retrieved, number of oocytes matured, and number of oocytes fertilized into respective categories. Not all eggs obtained were at the metaphase 2 stages and had to be matured in the incubator overnight and injected the following day. The results shown were to some extent anticipated as AMH has been used an indicator of oocyte reserve in previous studies (Yarde et al., 2013; Yao et al., 2015) whereas the resulting fertilized or transferred embryo's may be due to a chance process based on many various factors such as quality of oocyte and sperm.

The Chi-square test for Independence was performed to check whether there was an association between the number of oocytes fertilized and the AMH category (Table 2). A Chi-squared value of 18.5, degrees of freedom = 12, with a $p = 0.10$ was found. There was no statistically significant relationship between numbers of oocytes fertilized versus AMH category ($p > 0.05$).

Out of 22 patients, 43 embryos were transferred. Embryos were transferred depending on embryo development and the number of embryos obtained. Most patients in the high and normal categories resulted in a day 5 transfer, the Chi-squared test for independence of AMH and number of embryos transferred gave a Chi-squared value of 6.384 with df = 4 and $p$-value = 0.172 (Table 3), thus statistically no significant association between AMH and number of embryos transferred was observed. Whilst, Chi-square test for independence between the variables AMH and day of embryo transfer (Table 4) gave a Chi-square value of 14.117, 6 degrees of freedom and $p = 0.028$ indicating statistically significant relationship between AMH and day of embryo transfer ($p < 0.05$).

Pregnancy outcome and AMH category are as shown in Table 5. Out of twenty two cases in high category, 6 resulted in a positive pregnancy; 6 resulted in a positive outcome (6/12 = 50.0%) (Normal); while out of the 3 cases where the AMH was "Low to Normal", there were no pregnancies reported. The Chi-squared test for independence of AMH category and pregnancy outcome gave a Chi-Squared value of 0.502, 2 degrees of freedom

**Table 1  AMH distribution in blood samples.**

| Category | AMH Blood Level Concentration | Frequency | % Patients |
|---|---|---|---|
| High (often PCOS) | $\geq$ 3.0 ng/ml | 22 | 52.4 |
| Normal | $\geq$ 1.0 ng/ml | 17 | 40.5 |
| Low Normal Range | $\leq$ 0.3–0.9 ng/ml | 3 | 7.1 |

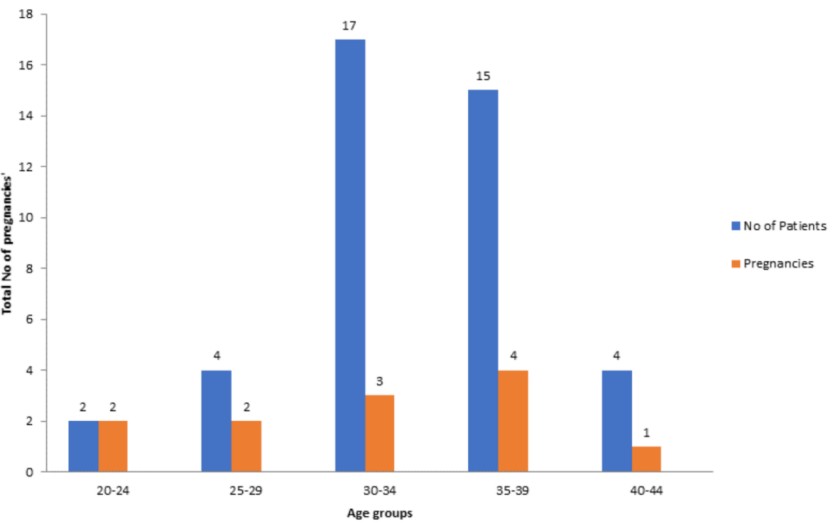

**Figure 1  Number of patients associated with pregnancies in relevant age groups.**

and $p = 0.778$. There was no statistically significant relationship between the pregnancy outcome and the AMH category ($p > 0.05$).

## Pearson correlation

Pearson correlation coefficients were calculated to determine if any statistical significance exists between AMH on a quantitative scale and age, $E_2$ and FSH (Table 6). The Pearson Correlation co-efficient of 0.151 indicates that a very weak positive relationship existence between $E_2$ and AMH, which is not statistically significant ($p = 0.341$). Furthermore, Pearson correlation coefficient between the AMH and age had a coefficient of $-0.028$ thus showing no statistical significance $p = 0.859$ ($p > 0.05$). The Pearson Correlation between AMH and FSH produced a coefficient of $-0.185$ thus indicating no statistical significance $p = 0.240$ ($p > 0.05$). Pearson correlation coefficient showed no significant association between AMH and number of oocytes ($p = 0.191$), number of mature oocytes ($p = 0.300$) and number of oocytes fertilized ($p = 0.146$). The number of oocytes, mature oocytes and oocytes fertilized all presented a no statistically significant correlation with AMH (0.206, 0.164, and 0.228, respectively).

## Logistic regression analysis

A logistic regression model was used to determine the possible predictor variables for the pregnancy outcome. The model was fitted to the data with the result of the pregnancy

**Table 2  Correlation between AMH and number of oocytes collected, matured, and fertilised during stimulation.**

| | | Total no. of oocytes collected | Total no. of oocytes matured | Total no. of oocytes fertilized | % oocytes collected | % oocytes matured | % oocytes fertilized |
|---|---|---|---|---|---|---|---|
| AMH category | High | 81 | 62 | 69 | 65.4% | 60% | 61.6% |
| | Normal | 38 | 36 | 38 | 30.6% | 35% | 33.9% |
| | Low to Normal | 5 | 5 | 5 | 4.0% | 5% | 4.5% |
| Total | | 124 | 103 | 112 | 100% | 100% | 100% |

**Table 3  Chi Square analysis of number of embryos transferred and AMH.**

| | Value | Degrees of Freedom (df) | Asymptotic Significance (2-sided) |
|---|---|---|---|
| Pearson Chi Square | 6.384[a] | 4 | .172 |
| Likelihood Ratio | 7.001 | 4 | .136 |
| N of Valid Cases | 42 | | |

**Notes.**
[a] Seven cells (77.8%) have expected count less than five. The minimum expected count is 0.08.

**Table 4  Chi Square analysis of AMH and day of embryo transfer.**

| | Value | Degrees of Freedom (df) | Asymptotic Significance (2-sided) |
|---|---|---|---|
| Pearson Chi Square | 14.117[a] | 6 | 0.028 |
| Likelihood Ratio | 6.432 | 6 | 0.377 |
| N of Valid Cases | 42 | | |

**Notes.**
[a] Ten cells (83.3%) have expected count less than five. The minimum expected count is 0.05.

namely, "Positive" or "Negative" as the binary dependent variable and age, $E_2$ LH, Basal FSH, Basal AMH and number of oocytes fertilized as independent variables (Table 7). As shown in Table 7, LH has $p = 0.042$ ($p < 0.05$) and $E_2$ has $p = 0.065$, which is statistically not significant at a 5% level. The SPSS output for the model is given in Table 1 (Data S1) signifying that overall, 73.8% of the cases were correctly classified, while $5/12 = 0.417$ or 41.7% of the positives were correctly classified, and 86.7% of the negative cases were correctly classified.

## Area under the curve

The ROC curves of the serum AMH concentrations and $E_2$ for the prediction of the clinical pregnancy are depicted in Fig. 2. The areas under the curves (AUC) for $E_2$ were 0.725 and for AMH (AUC = 0.497). $E_2$ is therefore a better single predictor of pregnancy outcome when compared to AMH. It has been shown that $E_2$ can better predict the number of oocytes obtained.

**Table 5  Pregnancy outcome and AMH category.**

| AMH category | Pregnancy result | | Total |
| --- | --- | --- | --- |
| | Negative | Positive | |
| High | 16 | 6 | 22 |
| Normal | 11 | 6 | 17 |
| Low Normal | 3 | 0 | 3 |
| Total | 30 | 12 | 42 |

**Table 6  Pearson correlation between basal AMH and $E_2$, Age and FSH and oocytes.**

| | | $E_2$ | Age | FSH | Number of oocytes | Number of mature oocytes | Number of oocytes fertilized |
| --- | --- | --- | --- | --- | --- | --- | --- |
| AMH | Pearson correlation coefficient | 0.151 | −0.028 | −0.185 | 0.206 | 0.164 | 0.228 |
| | Significance value (2-tailed) | 0.341 | 0.859 | 0.240 | 0.191 | 0.300 | 0.146 |
| | No. in the sample | 42 | 42 | 42 | 42 | 42 | 42 |
| Age | Pearson correlation | | | | −0.087 | −0.271 | |
| | Significance value (2-tailed) | | | | 0.583 | 0.082 | |
| | No. in the sample | | | | 42 | 42 | |

**Table 7  Logistic regression analysis of the variables for the prediction of pregnancy.**

| | B | Standard error (S.E.) | Wald | Degree of Freedom (df) | $p$-value | OR = Exp (B) |
| --- | --- | --- | --- | --- | --- | --- |
| $E_2$ | .001 | .000 | 3.396 | 1 | .065 | 1.001 |
| LH | −.556 | .273 | 4.144 | 1 | .042 | .574 |
| Basal AMH | −.335 | .239 | 1.967 | 1 | .161 | .715 |
| Age | −.146 | .091 | 2.593 | 1 | .107 | .864 |
| Non-Fertilized | .150 | .368 | .166 | 1 | .683 | 1.162 |
| Basal FSH | −.102 | .136 | .559 | 1 | .455 | .903 |
| Constant | 4.451 | 3.452 | 1.663 | 1 | .197 | 85.744 |

**Notes.**

$E_2$, Estrogen; LH, luteinizing hormone; AMH, anti-Müllerian hormone; FSH, follicle stimulating hormone.
LH has $p = 0.042$ ($p < 0.05$) and $E_2$ has $p = 0.065$, which is significant at a 10% level. In this logistic regression model, the remaining variables are not significant ($p$-values $> 0.10$).

## DISCUSSION

In the current study, we investigated the relationship between AMH levels and pregnancy outcomes in patients undergoing intra-cytoplasmic sperm injection.

### Correlation between basal AMH and $E_2$

Pearson analysis between $E_2$ and AMH presented a Pearson Correlation co-efficient of 0.151 with $p = 0.341$ ($p < 0.05$) which indicates that a weak significant relationship exists between $E_2$ and AMH. Most previous studies (*Ramalho de Carvalho et al., 2012*; *Ubaldi et al., 2005*) have shown a relationship between a raised basal $E_2$ level and a reduced ovarian response using different values to express elevated estrogen levels which replicated the findings in
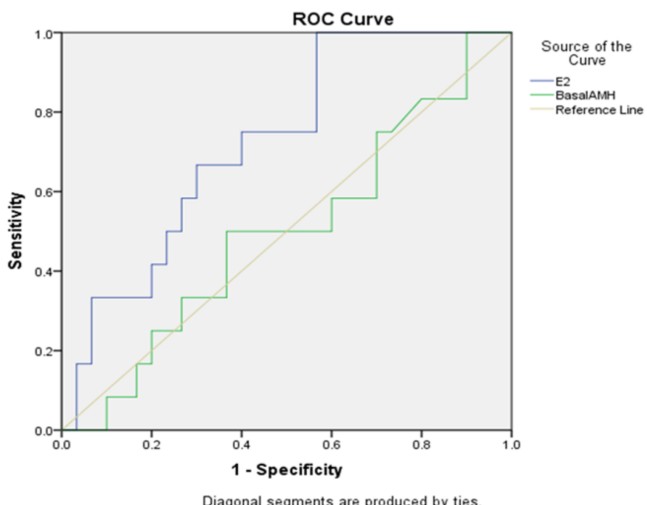

**Figure 2** Sensitivity and specificity of E₂ and AMH in predicting pregnancy.

this study therefore showing that a low AMH can result in low estrogen levels. Also, it can be concluded that a poor AMH value results in a poor ovarian reserve indicating follicles produced will not be correlated to a raised estrogen level, therefore indicating poor follicle growth, thus reducing the number of oocytes produced. However, it was determined that poor response to stimulus in IVF, indicative of a lower ovarian reserve, is associated with declined baseline serum AMH concentrations (*Van Rooij et al., 2004*). Consequently, when women have regular ovarian reserve and decent retort, disappointment of IVF must look for additional infertility reasons, e.g., male specific issue i.e., Y chromosome microdeletion. Furthermore, this conclusion is reinforced by the data of woman undergoing IVF which indicated that male factor infertility resulted in an unsuccessful cycle. Although E2 levels in these cases were above those of controls, they are still within the range of 25–100 pg/ml (*Sahmay et al., 2014*), suggesting that E2 single-handedly is not capable of predicting the female reproductive potential.

## Correlation between basal AMH and age

Pearson correlation between AMH and age (Table 6) presented a co-efficient of −0.028 thus displaying a weak, negative association with a $p = 0.859$ ($p > 0.05$). A stronger relationship between these two variables was expected as it is known that as age increases, AMH should decrease. This contrary association corresponds as reported by *Van Rooij et al. (2004)*, where serum AMH levels decline with age in normal women with proven fertility. Also, it is suggested that serum AMH is identified as the improved endocrine marker to measure the reproductive capability in advanced age.

## Correlation between basal AMH and FSH

Basal FSH is one of the primary endocrine markers presented into ART program. The Pearson correlation amid AMH and FSH (Table 6) had a coefficient of −0.185 thus displaying a weak, negative relationship and with a $p = 0.240$ ($p > 0.05$). This study

specifies a negative correlation, i.e., the higher the FSH the higher the chances the patient can present with a poor ovarian reserve and early menopause. This was strategic on the awareness that these women will respond well to ovarian stimulation while the basal FSH level is lesser at the beginning of the cycle. The outcomes of this study revealed that woman who were poor respondents or had a reduced ovarian reserve had a poor outcome and frequent testing is of no worth. Women who had a history of high FSH level must undergo treatment without further delay. By postponing treatment for these patients can be detrimental as they get older and fast approaching menopause (*Uzumcu & Zama, 2016*).

## Correlation between age and FSH and number of oocytes

The Pearson Correlation between FSH and age (Table 6) displayed no statistical significance, $p = 0.583$ ($p > 0.05$). For most of Pearson Correlation analysis, no significant relationships were found with most of the $p$-values, being greater than 0.05. This may be due to the small sample size used in this study of 42 patients. AMH being compared to age and number of oocytes showed a slightly negative correlation which is expected as it is shown in previous studies that AMH and number of oocytes decrease with maternal age (*Van Rooij et al., 2004*) and *Gobikrushanth et al. (2018)*. This inverse relationship is in agreement by *Van Rooij et al. (2004)*, who reported that serum AMH levels deteriorate with age in normal women with proven fertility. Additionally, serum AMH indicates the simplest endocrine marker to measure the age-related decline of reproductive competence. AMH levels, in our group who were high respondents were over 3.0 ng/ml, normal respondent over 1.0 ng/ml and low respondents found to be below 0.9 ng/ml. Oocytes were still recovered even with low AMH levels. Neither fertilization rate nor embryo quality can be assessed using basal AMH levels. This contrasts with the findings reported by *Vaegter et al. (2017)*, where embryos had superior morphology and cleavage performance in patients with AMH levels >2.7 ng/ml as compared with patients with values below this threshold.

## Correlation between AMH and the number of oocytes, number of mature oocytes and number of oocytes fertilized

Our second objective of the study was to examine if AMH levels affected oocyte quality. In this study, the Pearson Correlation test (Table 6) showed no significant relationship between AMH and number of oocytes ($p = 0.191$), several mature oocytes ($p = 0.300$) and number of oocytes fertilized ($p = 0.146$). The number of oocytes, mature oocytes and oocytes fertilized all showed a weak positive relationship to AMH (0.206, 0.164, and 0.228, respectively). These findings are in agreement with that reported by *La Marca & Sunkara (2014)*, *La Marca et al. (2010)* and *Dehghani, Tayebi & Asgharnia (2008)*, where mean amount of oocytes was lower in poor responding patients than in normal patients attending IVF programs. This therefore led to the inference that ovarian response can be regarded as a reflection of the ovarian reserve. The Chi-square test for Independence was done to determine whether there is an association between the number of oocytes collected and the AMH category (Table 8). A Chi-squared value of 21.246, degrees of freedom = 8, with a $p = 0.007$ was observed. There was a significant relationship between the numbers of oocytes collected versus AMH category ($p < 0.05$). The Chi-square test for

**Table 8  Chi Square analysis of Number of oocytes collected and AMH.**

|  | Value | Degrees of Freedom (df) | Asymptotic Significance (2-sided) |
|---|---|---|---|
| Pearson Chi Square | 21.246[a] | 8 | .007 |
| Likelihood Ratio | 21.317 | 8 | .006 |
| N of Valid Cases | 42 | | |

Notes.
[a] Thirteen cells (86.7%) have expected count less than five. The minimum expected count is 0.12.

**Table 9  Chi Square analysis of Number of oocytes fertilized and AMH.**

|  | Value | Degrees of Freedom (df) | Asymptotic Significance (2-sided) |
|---|---|---|---|
| Pearson Chi Square | 18.504[a] | 12 | .101 |
| Likelihood Ratio | 17.736 | 12 | .124 |
| N of Valid Cases | 42 | | |

Notes.
[a] Eighteen cells (85.7%) have expected count less than five. The minimum expected count is 0.02.

Independence was performed to see whether there is an association between the number of oocytes fertilized and the AMH category (Table 9). A Chi-squared value of 18.5, degrees of freedom = 12, with a $p = 0.10$ was found. There was thus no statistically significant relationship between the numbers of embryo's fertilized versus AMH category ($p > 0.05$). This is anticipated as AMH has been used an indicator of oocyte reserve in previous studies whereas the resulting fertilized or transferred embryo's may be due to a chance process based on many various factors such as the quality of the oocyte and sperm (*Yarde et al., 2013*). *Ebner et al. (2006)*, demonstrated that AMH serum levels were related with oocyte quality in stimulated cycles. The quality of the embryos was not assessed using baseline AMH which agrees with our findings. However, the fertilization rate was not correlated with the serum AMH which varied with the results of the present study.

## AMH category and positive pregnancies

Embryo quality has been suggested to be of paramount importance to predict the occurrence of pregnancy after IVF. In a regression model $E_2$ has a $p = 0.017$ ($p < 0.05$) and LH has a $p = 0.035$ ($p < 0.05$). Both variables are significant and age and basal AMH play a role in the pregnancy outcome and the model is thus adjusted for these two variables

In this study, AMH value for predicting pregnancy outcomes does not exist because oocyte quality is not accounted for by ovarian reserve markers. As demonstrated in this study the clinical pregnancy rate for patients 20–24 years was 100%, 25–29 was 50%, 30–34 years was 18%, 35–39 years was 27% and 40–44 years was 25% (Fig. 1). Patients presenting with a low AMH did not vary from those women presenting with higher AMH concentrations in same age group. A positive pregnancy outcome was logged across all age groups regardless of the AMH level. These results advocate that low ovarian reserve is not correlated with low oocyte quality in patients and the prediction remains the similar despite low AMH concentrations. *Kini et al. (2010)* stated the role of AMH in foreseeing

cumulative pregnancy outcome during IVF treatment. It was recognized that serum AMH concentration on day 6 of stimulation was suggestively higher in participants who resulted in an ongoing pregnancy in IVF compared to those who did not. Serum AMH is a suitable indicator of ovarian hyper-response. In a metanalysis study conducted by *Yao et al. (2015)* to evaluate role of serum AMH role in forecasting the pregnancy outcome in IVF/ICS, it was concluded that there is positive correlation between serum AMH and pregnancy. Nevertheless, association between serum AMH and non-pregnancy cannot be ruled out either.

## CONCLUSION

In conclusion, the outcomes of our investigations specify that AMH has established to be a valuable marker for ovarian reserve and might benefit woman who plan for pregnancy. AMH hormone seems to be the best endocrine marker, however, the valuable role of AMH and its role in ovarian function should be looked at in relation to the other markers to assess the decline of the ovarian pool. While appropriate reference values are being generated per age category and until the consequences of having a low or high AMH for one's age are being established, AMH should only be determined in the context of clinical studies. At present, the most important clinical role of AMH at this stage is to serve as a red flag for reduced ovarian reserve in women of reproductive age who must undergo further diagnostics. As per the study conducted, we can infer that AMH can accurately predict ovarian reserve but cannot predict the oocyte quality or a positive pregnancy outcome. The more oocytes obtained, increases a patient's chance of more viable embryos and therefore, improving chances of a healthy pregnancy and ultimately a live birth. Further research on the implication of varying levels of AMH within the follicular fluid may be representative as an indicator of "quality" in addition to the number of growing follicles.

## STUDY LIMITATION

A noteworthy restraint of the current study was the lack of antral follicle count (AFC) at time of oocyte collection.

### Funding
The authors received no funding for this work.

### Competing Interests
The authors declare there are no competing interests.

### Author Contributions
- Shalini Umarsingh, Jamila Khatoon Adam and Suresh Babu Naidu Krishna conceived and designed the experiments, performed the experiments, analyzed the data, prepared figures and/or tables, authored or reviewed drafts of the paper, and approved the final draft.

## Human Ethics

The following information was supplied relating to ethical approvals (i.e., approving body and any reference numbers):

This study was approved by Ethical Committee of the Durban University of Technology (Project reference 128/16) and Research Committee, C.A.R.E. Clinic, Durban, South Africa and was performed in accordance with the Helsinki Declaration of 1975 (as revised in 1983).

## Data Availability

Raw data are available as Supplemental Files.

## Supplemental Information

Supplemental information for this article can be found online at http://dx.doi.org/10.7717/peerj.10390#supplemental-information.

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
