# Peer review of "The relationship between anti-Müllerian hormone (AMH) levels and pregnancy outcomes in patients undergoing assisted reproductive techniques (ART)"

_PeerJ, doi:10.7717/peerj.10390_

## Round 0.1 · original submission · Minor Revisions

Both reviewers find the manuscript well written, while some details need to be clarified, such as selection of patients, dose of gonadotropins, and infertility causes.

Another issue is that, as both reviewers point out, the sample size of 42 analyzable patients seems a little too small for a study with correlative analysis and may be underpowered. So, authors need to clearly address this limitation in the revised manuscript.

Authors are also suggested to modify or clearly explain those terms, such as “non-parametric Pearson correlation” and “weak correlations”.

See reviewers’ comments for more information.

·

Basic reporting

The article is well written and well divided.
Nevertheless, the article does not bring anything new to the subject that has been widely discussed in the last years.
The objective of the study was to investigate the relationship between Anti-Müllerian
hormone levels and pregnancy outcomes in patients undergoing in-vitro fertilization or intra-cytoplasmic sperm injection (ICSI), although, it does not match (not reflect) the title of the manuscript.

Experimental design

The biggest problem is the selection of patients.

The age of the patients (> 40 years) can in itself negatively influence the assisted reproduction outcomes, ends up becoming a study bias.

What was the total dose of gonadotropins used? Did all patients use the same amount, regardless of age? Outcomes should not be compared if different protocols or different amounts of gonadotropins were used.

What were the infertility causes of the patients? Ovulatory factors, endometriosis and severe male factor also negatively influence reproduction outcomes, influence AMH serum values and therefore are biases in the study. Did all patients have both ovaries?

The number of patients is another very important issue. The analysis of 50 or 42 patients is insufficient to analyze whether a serum measurement can be used as a biomarker.

Validity of the findings

It is very difficult to assess whether the results are valid in the face of so many biases found at the study.

You need to adjust the number of patients and especially the patient inclusion criteria.

Reviewer 2 ·

Basic reporting

Clarity is needed on lines 61-67. Is the intent in this paragraph to indicate that GnRH agonist induced gonadotropin levels correlate with antral follicle counts? Is this referring purely to endogenous gonadotropin in circulation?

Please include the trade names and catalogue numbers for the immunoassays for the purpose of comparisons to other similar studies in the literature.

Experimental design

The sample size of 42 analysable patients is very small for a study with correlative analysis and may be underpowered.

Please clarify what is meant by “non-parametric Pearson correlation”. Typically, Pearson correlation is parametric and Spearman-rank correlation is non-parametric.

Validity of the findings

Lines 167-174 if the p-value of the correlative analysis is greater than 0.05, then it indicates that there is no evidence of an association and that the slope of the line has arisen through random variation within the data. It is not appropriate to refer to these results as “weak correlations” or “weak negative correlations”.

On lines 120-121 the level of significance is indicated to be p < 0.05 but on line 181 it refers to a p-value of 0.065 as significant at the 10% level. This lack of consistency should be addressed.

Additional comments

The study by Umarsingh et al investigates whether AMH levels predict pregnancy outcomes in patients undergoing ICSI. Consistent with other studies the results show that AMH is not correlated with pregnancy outcome. At 42 participants, this is a very small study for correlative analysis and may explain why a large number of non-significant results were obtained, as there may have been insufficient statistical power. However, it is important that the authors do not refer to correlative analysis results with non-significant p-values as weak correlations (see comments below).

---

## Round 0.2 · Minor Revisions

Certain comments and critiques from the reviewer had not been well addressed. See reviewer's comments for details.

Reviewer 2 ·

Basic reporting

1. Reviewer’s original comment: 1. Clarity is needed on lines 61-67. Is the intent in this paragraph to indicate that GnRH agonist induced gonadotropin levels correlate with antral follicle counts? Is this referring purely to endogenous gonadotropin in circulation?

Author’s response: Yes, it refers to endogenous gonadotropin circulation. As woman age, they have fewer oocytes (primordial follicles) remaining and they have fewer antral follicles. Antral follicle counts are a good predictor of mature follicles that can be stimulated in the ovary when stimulation medication is administered. Counting antral follicles using an ultrasound can be a subjective process. An ideal antral follicle count depends on the age of the woman; older women are not expected to have the same antral follicle

Reviewer’s response: Clarity is still needed in this paragraph, as no changes have been made. The intent of the authors is not clear and it needs to be redrafted. The authors response mentions antral and mature follicle counts which do not feature in the paragraph. It is not clear whether the authors are referring to endogenous gonadotropin or gonadotropin administered during stimulation cycles.



2. Reviewer’s original comment: Please include the trade names and catalogue numbers for the immunoassays for the purpose of comparisons to other similar studies in the literature.

Author’s response: The trade names of the compounds/drugs procured are listed in manuscript in Materials and Methods. However, for clarity the catalogue numbers are as follows:
• B84493 Access Sensitive Estradiol Reagent, 100 Determinations, 2 x 50 tests.
• B13127 Access AMH (Anti-Mullerian Hormone) Reagent, 100 Determinations, 2 x 50 Tests.
• 33520 Access hFSH Reagent, 100 Determinations, 2 x 50 tests

Reviewer’s response: These reagent details listed above have not been included in the methods section of the manuscript. The readers of the final article will need to know where the reagents came from if they want to replicate this study or compare it with others. The methods section of the manuscript refers to a manual 96-well plate format AMH Gen II ELISA (referencing Kumar et al. 2010) whereas the regent listed above is for an automated immunoassay. These two immunoassays have different properties. Which one was used? (state this in the manuscript)

Experimental design

Reviewer’s original comment: Please clarify what is meant by “non-parametric Pearson correlation”. Typically, Pearson correlation is parametric and Spearman-rank correlation is non-parametric.

Author’s response: Yes, we agree with reviewer and amended the manuscript likewise to Pearson correlation to avoid the confusion (L 116) with the readers.

Reviewer’s response: The manuscript methods section still reads “non-parametric Pearson correlation”. Pearson correlation is parametric. The authors have not made the change they claim above.

Validity of the findings

Reviewer’s original comment: 1. Lines 167-174 if the p-value of the correlative analysis is greater than 0.05, then it indicates that there is no evidence of an association and that the slope of the line has arisen through random variation within the data. It is not appropriate to refer to these results as “weak correlations” or “weak negative correlations”.

Author’s response: Yes, we agree with reviewers’ comment and completed necessary amendments as “not statistically significant” in the text (L 178)-red font

Reviewer’s response: If the p-value is >0.05 then there is no evidence of correlation. The manuscript still refers to these results as “correlations” in the results section. The requested changes have not been made, nor addressed with a rebuttal.

---

## Round 0.3 · accepted · Accept

Questions have been addressed.

Reviewer 2 ·

Basic reporting

-

Experimental design

-

Validity of the findings

-

Additional comments

My prior concerns have all been addressed